

# Predicting dominant terrestrial biomes at a global scale using machine learning algorithms, climate variable indices, and extreme climate indices

Hisashi SATO[1, 2]

[1] Research Institute for Global Change, Japan Agency for Marine-Earth Science and Technology (JAMSTEC), Yokohama, 236-0001, Japan
[2] Department of Ecosystem Studies, Graduate School of Agricultural and Life Sciences, The University of Tokyo, Tokyo, 113-8657, Japan

*Correspondence to*: Hisashi SATO (hsatoscb@gmail.com)

**Abstract.** Several methods have been proposed for modelling global biome distribution. Climate data are typically summarised in terms of a few climate indices. However, with the recent advancement of machine learning algorithms, such summarisation is no longer required. Extreme climate events such as intense droughts and very low temperatures cannot be captured by monthly mean climate data, which may limit the applicability of biome boundaries. In this study, I assessed the influences of machine learning algorithms, climate variable indices, and extreme climate indices on the accuracy and robustness of global biome modelling. I found that the random forest and convolutional neural network algorithms produced highly accurate models for reconstructing the global biome distribution. However, the convolutional neural network algorithm was preferable, because the random forest algorithm substantially overfit the training data relative to the other machine learning algorithms examined. Including indexed climate data slightly reduced model accuracy, whereas including extreme climate data slightly improved it. However, there were significant deviations in the distribution of values between the observed and predicted climate when extreme climate data was included; this fatally reduced the robustness of the models, which were evaluated in terms of prediction consistency. Therefore, I recommend that extreme climate data not be considered in global-scale biome prediction applications.



## 1. Introduction

A biome is a major regional ecological community characterised by distinctive life forms and principal plant species (Lincoln et al., 1998). Biome distributions are useful for estimating land potential and raising public awareness about land change (reviewed in Hengl et al. (2018)). At the global scale, climate conditions largely determine biome distribution (Adams, 2010; Prentice et al., 1992), and biome distribution interacts with climate through biophysical and biochemical pathways (Pitman, 2003). Thus, biome distributions may also be applied in climate projection.

To date, several methods have been proposed for modelling biome distribution (Sato and Ise, 2022). In these models, climate data like monthly mean temperature and monthly precipitation are typically summarised as smaller numbers of climate indices such as annual precipitation and coldest month mean temperature. However, with the recent advancement of machine learning algorithms such as random forest (RF), restrictions on the amount of data used in model building have been relaxed, and it is no longer essential to summarise environmental data within indices. For example, Hengl et al. (2018) used 160 environmental variables including soil and topography, as well as non-indexed climate variables such as monthly precipitation and monthly average temperature to construct an empirical model of biome distribution using machine learning algorithms. However, increasing the number of variables in the model entails costs such as lower model adaptability and higher computational demand; therefore, it is still important to limit the number of variables included in the model.

From the perspective of plant physiology and ecology, the intensity of extreme climate events such as severe droughts and rare low-temperature incidents is a significant factor limiting biome boundaries (reviewed in Beigaite et al., 2022). Including extreme climate indices in addition to non-extreme climate variables has been reported to increase the accuracy of decision tree models (Beigaite et al., 2022).

In this study, I evaluated the accuracy of four machine learning algorithms in a global biome distribution model based on current climate characteristics. I also assessed the influence of including extreme climate indices or converting monthly precipitation and average air temperature (24 variables) into 16 climatic indices on model accuracy. To



explore how the resulting models responded to climatic conditions beyond the training
data, I applied them to forecast biome distributions for future climatic conditions (2060–
2080) and compared their outputs.
**2. Methods**
**2.1 Biome data**
I used potential natural vegetation (PNV) compiled by Beigaite et al. (2022) to develop
decision tree-based models of global PNV distribution
(https://github.com/ritabei/dominant-natural-vegetation, accessed 20 June, 2022). The
original PNV data were obtained from the Moderate Resolution Imaging
Spectroradiometer (MODIS) MCD12C1 land cover product in 2001
(https://doi.org/10.5067/MODIS/MCD12C1.006). The MCD12C product contains three
land cover classifications, among which Beigaite et al. (2022) used the International
Geosphere Biosphere Programme (IGBP) land cover classification, which is primarily
based on supervised learning classification of MODIS Terra and Aqua reflectance data
(Friedl et al., 2010). The MCD12C1 product contains percent cover for 17 IGBP classes
(Loveland and Belward, 1997) in each grid cell at a resolution of 0.05°. Beigaite et al.
(2022) resampled the MCD12C1 data to 50 km × 50 km grids and extracted the dominant
natural vegetation with the highest fraction in each grid cell. Among the original 17
categories, only 13 (natural vegetation) were used in this study (Figure 1, Table SI 1).
Thus, grid cells with 100% human activity or water cover, or a combination of both, were
eliminated from the analysis. I also ignored the continent of Antarctica. Ultimately,
52,297 grid cells were included in the analysis.
**2.2 Climate data**
This study used four climate datasets: averaged monthly air temperature and precipitation
(*Ave*, 24 variables), averaged monthly climate indices (*AveI*, 16 variables), climate
extreme indices representing extreme conditions on a daily scale such as the maximum
length of a dry spell (*CEI*, 27 variables), and a subset of *CEI* (*CEI$_{part}$*, 21 variables). The
variables included in *AveI* and *CEI* are listed in Tables SI 2 and SI 3, respectively. Figures





SI 1–3 show the present (1970–2000) and future (2061–2080) distributions of *Ave*, *AveI*,
and *CEI*, respectively. Among all climatic variables used in this study, only six variables
in the *CEI* dataset (Tn10p, Tx10p, Tn90p, Tx90p, WSDI, and CSDI) had completely
separate variable distributions between the present and future. Another indexed extreme
climate dataset, *CEI$_{part}$*, was constructed by excluding these variables from the *CEI*
dataset.
*Ave* data were obtained from the WorldClim 2.1 product (released January 2020; Fick
and Hijmans (2017)), which represents average monthly air temperature and precipitation
data for 1970–2000. The original WorldClim 2.1 product was downloaded
(http://worldclim.org, accessed 01 July, 2022) at a spatial resolution of 10 min, and
resampled to 50 km × 50 km grids using the nearest-neighbour method. *AveI* was released
by Beigaite et al. (2022), summarising WorldClim 2.1 properties in terms of annual means
(e.g., BIO1 and BIO12), seasonality (e.g., BIO4, BIO7, and BIO15), and limiting
environmental factors to a monthly scale (e.g., BIO5, BIO6, and BIO14).
The *CEI* product was released by Beigaite et al. (2022) using the CLIMDEX climate
extremes    index    (Sillmann,    2013    #3057@@author-year;    https://climate-
modelling.canada.ca/climatemodeldata/climdex/). CLIMDEX comprises four datasets
that were derived from different reanalysis datasets. Among these, Beigaite et al. (2022)
used a dataset calculated from the ERA-Interim reanalysis dataset, which accurately
reproduces observed climate extremes (Donat et al., 2014). *CEI* data derived from the
ERA-Interim reanalysis dataset covers 32 years (1979–2010). For each grid, multi-year
*CEI* values were averaged; multi-year averages of extreme indices are commonly used to
represent averaged extreme conditions in the past and future (Seneviratne and Hauser,
2020; Sillmann et al., 2013a). The original resolution of the *CEI* data was 1.5° × 1.5°;
they were transformed onto 10 min × 10 min grids through conservative interpolation and
then resampled to 50 km × 50 km grids using nearest-neighbour interpolation.
For    future    climate    condition    projections    (2061–2080),    I    used    BIOCLIM
(http://www.worldclim.com/cmip5_10m) and extreme climate variables (Sillmann et al.
(2013b);    https://crd-data-donnees-rdc.ec.gc.ca/CCCMA/products/CLIMDEX/CMIP5/)
derived from future climate projections of the International Panel on Climate Change





(IPCC) Coupled Model Intercomparison Project Phase 5 (CMIP5). I used only one future
scenario, Representative Concentration Pathway (RCP) 8.5. In this study, RCPs represent
atmospheric greenhouse gas (GHG) concentration forecasts adopted by the IPCC for its
Fifth Assessment Report (AR5) in 2014; RCP8.5 assumes that global annual GHG
emissions will continue to rise throughout the 21$^{st}$ century, resulting in 758 ppm of
atmospheric $CO_2$ by 2080 (IPCC, 2013). All data were based on ensemble means of 11
models participating in CMIP5 and averaged over 2061–2080.
**2.3 Machine learning algorithms**
I employed four machine learning algorithms: RF (Breiman, 2001), support vector
machine (SVM) (Cortes and Vapnik, 1995), naive Bayes classifier (NV) (Langley et al.,
1992), and LeNet convolutional neural network (CNN). RF, SVM, and NV algorithms
are commonly used to develop supervised learning models for classification. I
implemented and evaluated these algorithms using the *randomForest*, *ksvm*, and
*naiveBayes* packages in R v3.3.3 (R-Core-Team, 2018). I used the default model
parameters for simplicity and to prevent potential overfitting, i.e., training the model too
closely to a particular dataset, thereby creating a model that might fail to fit additional
data or reliably predict future observations.
CNN algorithms are more complex than the others included in this study. They are
typically applied to analyse visual imagery, and have been successfully adapted for
species distribution modelling at regional (Benkendorf and Hawkins, 2020; Botella et al.,
2018) and global scales (Sato and Ise, 2022). I follow Sato and Ise (2022) in training our
CNN with graphical images as input variables representing climatic conditions.
In contrast to Beigaite et al. (2022), I did not include a decision tree algorithm in our
study. Although decision trees rapidly provide interpretable boundary conditions for the
distribution of a given output variable, they are generally inferior to the algorithms
explored in this study in terms of reconstruction accuracy. The RF algorithm is an
ensemble of decision tree algorithms, which I anticipated would provide higher model
accuracy.





**2.4 Data analysis**

To separate the influences of climate data and extreme climate indices on PNV model performance, I compared the learning performance of six climate dataset combinations: *Ave*, *Ave + CEI*, *Ave + CEIpart*, *AveI*, *AveI + CEI*, and *AveI + CEIpart*. Four machine learning algorithms were applied for each climate dataset combination, resulting in 24 models.

For each model, 25% of all 52,297 grids were randomly selected and used for training. I determined the test accuracy of each model by calculating the ratio of correct answers when the model was applied to the remaining 75% of grids. I also determined the training accuracy of each model by calculating the ratio of correct answers when the model was applied to the training data itself. Generally, training accuracy scores were expected to be higher than test accuracy scores due to overfitting (Leinweber, 2007); thus, an overfitting score was calculated as training accuracy minus test accuracy. Ten experiments were conducted for each model, and their averages were compared among models.

**3. Results**

Irrespective of the training datasets, all models except NV reconstructed global PNV precisely (Figs. 1 and SI4–7). The ranges of test accuracy values were 80.1%–81.4%, 74.6%–78.0%, 44.2%–50.1%, and 77.1%–82.0% for the RF, SVM, NV, and CNN models, respectively (Table 1). All accuracy values were > 17.8%, in which all grids were assumed to be the most frequent PNV, grassland (Table SI1). All accuracy values except for NV were > 49%, in which all grids at the same latitude were assigned the most frequent PNV at that latitude.

The low test accuracy of the NV model was caused by an overestimation of areas dominated by boreal forest, tropical rainforest, and deciduous broadleaf forest (Fig. 4), whereas the other models tended to show grid discrepancies along PNV boundaries (Figs. 2, 3, and 5). This trend corresponds to that of observation-based biome distributions being fragmented along PNV boundaries (Fig. 1). In contrast, model-reconstructed biome distributions have more continuous structures (Figs. SI4–7). From further analysis and discussion, I excluded the NV model due to its poor performance.

The models shared common test accuracy patterns in response to input data (Table 1).
The summarizing climate data into indices decreased model test accuracy, with *AveI* −
*Ave* accuracy results of −1.1%, −1.8%, and −2.0% for RF, SVM, and CNN, respectively.
The inclusion of extreme climate indices (*CEI*) increased test accuracy, with (*Ave* + *CEI*)
− *Ave* accuracy results of 0.2%, 1.6%, and 1.0% for RF, SVM, and CNN, respectively,
and (*AveI* + *CEI*) − *AveI* accuracy results of 1.1%, 3.1%, and 2.8% for RF, SVM, and
CNN, respectively. Replacing *CEI* with *CEI*$_{part}$ in these comparisons revealed no
consistent trend in test accuracy, with negligible change for RF (0.1% vs. 0.1%), a
decrease for SVM (−0.3% vs. −0.8%), and an increase for CNN (1.7% vs. 2.1%).
Training accuracy rates consistently exceeded test accuracy rates for all combinations of
models and datasets (Tables 1 and 2), resulting in a positive overfitting score, defined as
training accuracy minus test accuracy (Table 3). The RF model always had 100% training
accuracy, resulting in high overfitting scores of 18.6%–20.0%. The overfitting scores of
the other models were much lower, at 1.38%–2.05% for SVM and 0.75%–2.17% for
CNN.
All models reconstructed highly coincident PNV distributions under current climatic
conditions, irrespective of the training datasets (accuracy, 70.1%–86.4%, Table 4). For
any combinations of models, datasets provide only slight differences in the
correspondence of PNV reconstructions: in comparing RF and SVM, which provides the
closest PNV distributions (accuracy, 84.5%–86.4%), the difference is less than 2.0%,
while in comparing SVM and CNN, which provides the farthest PNV distributions
(accuracy, 70.1%–72.4%), the difference is less than 2.3%.
When the trained models were adapted for a future climate, i.e., climate conditions
beyond the training data, there were much larger differences between PNV distributions
produced by different combinations of models and datasets (accuracy, 4.1%–82.8%,
Table 5), with larger discrepancies in PNV maps constructed by models trained with *CEI*
datasets (Figs. 6–9). SVM models trained with the *CEI* dataset output only evergreen
broadleaf forest (Fig. SI9c, d), whereas CNN models output maps with abundant
grassland and savanna (Fig. SI11c, d). Replacing *CEI* data with *CEI*$_{part}$ data amended
these extreme outputs (Figs. SI9e, f and 11e, f). Excluding models trained with the NV



algorithm and *CEI* dataset produced highly coincident PNV distributions under a future
climate (accuracy, 51.7%–82.8%, Table 5).

## 4. Discussion

Irrespective of the input dataset combination, the RF and CNN algorithms provided more
accurate global PNV models than did the SVM and NV algorithms. Hengl et al. (2018)
also found that RF consistently outperformed other machine learning algorithms,
including neural networks. In their study, a stack of 160 global maps representing
biophysical conditions over the terrestrial surface, including atmospheric, climatic, relief,
and lithologic variables, were used as explanatory variables to predict 20 biome classes
in the BIOME 6000 dataset (http://doi.org/10.17864/1947.99). Although a direct
comparison with the findings of the current study is impossible, this previous report
supports RF as a robust machine learning algorithm for reconstructing biome maps. The
present study is the first to compare the results of a CNN algorithm adapted for biome
modelling (Sato and Ise, 2022) to those of biome models based on other machine learning
algorithms; this CNN showed comparable performance to an RF.
I found that RF and CNN algorithms had similar test accuracy rates. However, the CNN
is preferable because RF produced much higher overfitting scores than any other machine
learning algorithm examined in this study. Overfitting is an inevitable risk associated with
empirical models (Leinweber, 2007). Fourcade et al. (2018) demonstrated an extreme
example of pseudo-predicting variables (randomly chosen classical paintings) increasing
the accuracy of species distribution modelling; these models sometimes had even higher
evaluation scores than models trained with relevant environmental variables. To avoid
overfitting or employing pseudo-predicting variables, Fourcade et al. (2018) suggested
expending more effort in cross-validation and ensuring the selection of the most important
predictors. I followed this suggestion in our analysis.
The climate index data used in this study reduced the number of variables by two thirds
(from 24 to 16). However, it reduced model accuracy only slightly (−1.1%, −1.8%, and
−2.0% for RF, SVM, and CNN, respectively), demonstrating that the typical climate
indices used in this study adequately extracted essential climate information relevant to



global biome distribution. Nevertheless, indexing has no particular merit in building
machine learning-based invisible models, whereas it is essential in building visible
models such as decision trees (Beigaite et al., 2022).
Adding extreme climate data improved test accuracy rates slightly but it can fatally reduce
model robustness, which was defined as the consistency of model prediction under
forecast climate conditions. This outcome was caused by six *CEI* variables with
distributions that were entirely distinct from the training data, demonstrating the need to
assess the distributions of training and predicting variables when building empirical
models. Because the slight improvement in test accuracy obtained by including extreme
climate data was insufficient to compensate the loss of model robustness, I recommend
that extreme climate data not be included in models predicting global biome distribution
at the geographical resolution employed in this study (0.5°).
There is clear evidence that climate extremes control plant demographic processes such
as growth (Jolly et al., 2005; Ciais et al., 2005), regeneration (Ibanez et al., 2007), and
mortality (Villalba and Veblen, 1998; Bigler et al., 2006), all of which influence plant
species distributions. However, it does not follow that extreme climate data should always
be considered to improve biome map reconstruction, because mean climatic values are
tightly correlated with extreme climatic variables. Even indexed climate variables
adequately extracted this correlated information in the present study, as shown by the
slight differences in test accuracy rates between *Ave* and *AveI* (< 1.8%), and between *Ave*
+ *CEI$_{part}$* and *AveI* + *CEI$_{part}$* (< 0.8%), except for NV models (Table 1). However, at local
and species levels, extreme climate may be a more critical predictor; Zimmermann et al.
(2009) revealed that complementing mean climate predictors with variables representing
climate extremes improves the predictive power of species distribution models.
A crucial disadvantage of the climatic envelope approach is that extrapolating current
correlations between climate and biome distributions into the future may lead to seriously
biased predictions. Thus, strong model performance under the present climate does not
guarantee similar performance under a new set of climatic conditions that may occur in
the future. However, no models except those trained with the NV algorithm and the *CEI*
dataset showed apparent expansions of PNV uncertainty under projected climatic



conditions. This result suggests that robust models can be developed beyond the training
data if the machine learning algorithms and climatic variables are carefully selected. The
climatic envelope approach has other limitations and disadvantages. For example, it
ignores time lags between climate change and vegetation change, changes in atmospheric
$CO_2$, and human land use change (discussed in Sato and Ise (2022)). However, the
climatic envelope is helpful for various adaptations, including benchmarking dynamic
global vegetation models (Fisher et al., 2018).
**5. Conclusion**
Models constructed based on RF and CNN algorithms provided accurate, robust global
PNV models. Despite its slightly higher accuracy, the RF model tended to overfit the
training data, leading to dramatically lower robustness; thus, the CNN model is
preferable. The inclusion of climate data indices has no particular merit in developing
"non-transparent" models, and only slightly reduced model accuracy. Extreme climate
data improves model accuracy and robustness only if the distributions of climate variables
are moderately within the range of the training data. Therefore, it is safer not to include
extreme climate indices.
**Data availability**
All data required to reproduce the analyses described herein are publicly available at the
following URL/DOI: https://doi.org/10.5281/zenodo.8113935.
**Acknowledgements**
This work was funded by the Arctic Challenge for Sustainability II (ArCS II) [Program
Grant Number JPMXD1420318865]. The authors declare no conflicts of interest.

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



**Tables**

Table 1. Average test accuracy rates ± standard deviation (%; n = 10) for models based on four machine learning algorithms: random forest (RF), support vector machine (SVM), Naive Bayes classifier (NV), and convolutional neural network (CNN). Input variable abbreviations: *Ave*, averaged monthly air temperature and precipitation; *AveI*, averaged monthly climate indices; *CEI*, climate extreme indices; and $CEI_{part}$, a subset of *CEI*.

| Input variable combinations | RF | SVM | NV | CNN |
|---|---|---|---|---|
| *Ave* | 81.2 ± 0.21 | 76.4 ± 0.15 | 46.7 ± 0.84 | 79.1 ± 0.15 |
| *Ave + CEI* | 81.4 ± 0.20 | 78.0 ± 0.15 | 45.2 ± 1.20 | 80.1 ± 0.12 |
| *Ave + CEI$_{part}$* | 81.5 ± 0.22 | 77.7 ± 0.19 | 44.2 ± 1.24 | 81.8 ± 0.30 |
| *AveI* | 80.1 ± 0.22 | 74.6 ± 0.12 | 50.1 ± 0.88 | 77.1 ± 0.18 |
| *AveI + CEI* | 81.2 ± 0.21 | 77.7 ± 0.19 | 44.6 ± 1.82 | 79.9 ± 0.16 |
| *AveI + CEI$_{part}$* | 81.3 ± 0.18 | 76.9 ± 0.16 | 43.3 ± 2.26 | 82.0 ± 0.31 |





Table 2.
As in Table 1, but for training accuracy rate ± standard deviation (%, n = 10).

| Input variable combinations | RF | SVM | NV | CNN |
|---|---|---|---|---|
| *Ave* | 100.0 ± 0.00 | 77.8 ± 0.28 | 46.8 ± 0.70 | 81.1 ± 0.90 |
| *Ave + CEI* | 100.0 ± 0.00 | 79.9 ± 0.25 | 45.3 ± 1.02 | 81.9 ± 0.94 |
| *Ave + CEI$_{part}$* | 100.0 ± 0.00 | 79.5 ± 0.32 | 44.4 ± 1.08 | 83.0 ± 0.44 |
| *AveI* | 100.0 ± 0.00 | 76.1 ± 0.33 | 50.5 ± 0.97 | 78.3 ± 1.02 |
| *AveI + CEI* | 100.0 ± 0.00 | 79.8 ± 0.14 | 44.7 ± 1.74 | 82.1 ± 0.90 |
| *AveI + CEI$_{part}$* | 100.0 ± 0.00 | 78.5 ± 0.38 | 43.5 ± 2.14 | 82.9 ± 0.48 |


Table 3.
As in Table 1, but for overfitting scores ± standard deviation (%, n = 10).

| | RF | SVM | NV | CNN |
|---|---|---|---|---|
| *Ave* | 18.9 ± 0.21 | 1.38 ± 0.30 | 0.11 ± 0.40 | 2.05 ± 0.99 |
| *Ave + CEI* | 18.7 ± 0.20 | 1.92 ± 0.30 | 0.15 ± 0.36 | 1.78 ± 0.91 |
| *Ave + CEI$_{part}$* | 18.6 ± 0.22 | 1.83 ± 0.43 | 0.14 ± 0.32 | 0.75 ± 0.62 |
| *AveI* | 20.0 ± 0.22 | 1.53 ± 0.39 | 0.33 ± 0.59 | 1.19 ± 1.03 |
| *AveI + CEI* | 18.8 ± 0.21 | 2.05 ± 0.29 | 0.15 ± 0.49 | 2.17 ± 0.86 |
| *AveI + CEI$_{part}$* | 18.8 ± 0.18 | 1.91 ± 0.46 | 0.16 ± 0.43 | 1.06 ± 0.57 |






Table 4. Degree of coincidence (%) in pairwise comparisons of simulated potential
natural vegetation (PNV) under the current climate. Asterisks indicate the exclusion of
models trained with the naive Bayes classifier.

|  | RF vs SVM | RF vs NV | RF vs CNN | SVM vs NV | SVM vs CNN | NV vs CNN |
|---|---|---|---|---|---|---|
| *Ave* | 85.6* | 49.4 | 70.8* | 50.9 | 70.1* | 33.5 |
| *Ave + CEI* | 86.4* | 47.8 | 71.5* | 49.0 | 72.3* | 32.0 |
| *Ave + CEI$_{part}$* | 86.3* | 46.1 | 74.1* | 46.9 | 70.6* | 30.4 |
| *AveI* | 84.4* | 53.8 | 70.8* | 56.7 | 71.9* | 38.8 |
| *AveI + CEI* | 86.1* | 46.9 | 70.9* | 48.1 | 72.4* | 31.2 |
| *AveI + CEI$_{part}$* | 85.5* | 45.0 | 73.5* | 46.0 | 70.7* | 29.2 |


Table 5. Degree of coincidence (%) in pairwise comparisons of simulated potential
natural vegetation (PNV) under a Representative Concentration Pathway 8.5 (RCP8.5)
climate scenario. Asterisks indicate the exclusion of models trained with the naive
Bayes classifier or including climate extreme indices as input data.

|  | RF vs SVM | RF vs NV | RF vs CNN | SVM vs NV | SVM vs CNN | NV vs CNN |
|---|---|---|---|---|---|---|
| *Ave* | 78.6* | 46.0 | 56.3* | 52.8 | 65.4* | 35.8 |
| *Ave + CEI* | 3.4 | 21.0 | 43.4 | 20.2 | 1.6 | 2.8 |
| *Ave + CEI$_{part}$* | 82.8* | 45.7 | 63.1* | 51.2 | 51.7* | 30.6 |
| *AveI* | 81.2* | 51.5 | 60.8* | 56.9 | 65.4* | 37.7 |
| *AveI + CEI* | 4.1 | 22.0 | 56.8 | 17.5 | 5.1 | 10.2 |
| *AveI + CEI$_{part}$* | 82.0* | 44.9 | 66.3* | 49.4 | 66.0* | 30.0 |






**Figures**

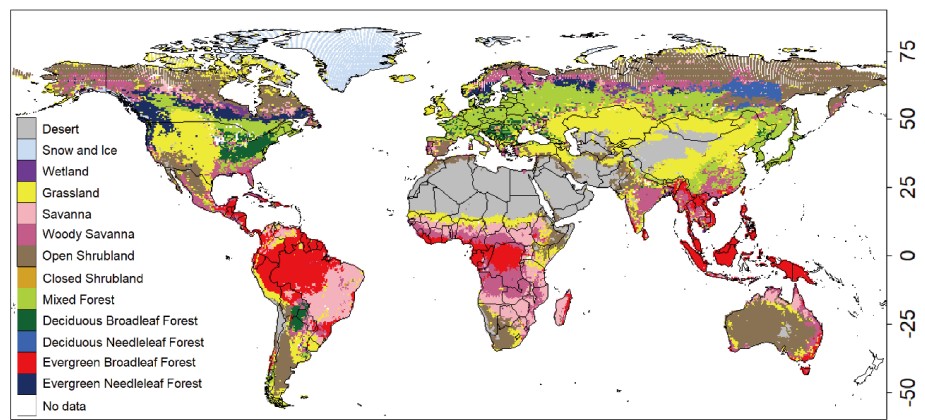

Figure 1

Distribution of observation-based potential natural vegetation (PNV) data, which were used to train machine learning-based models in this study.



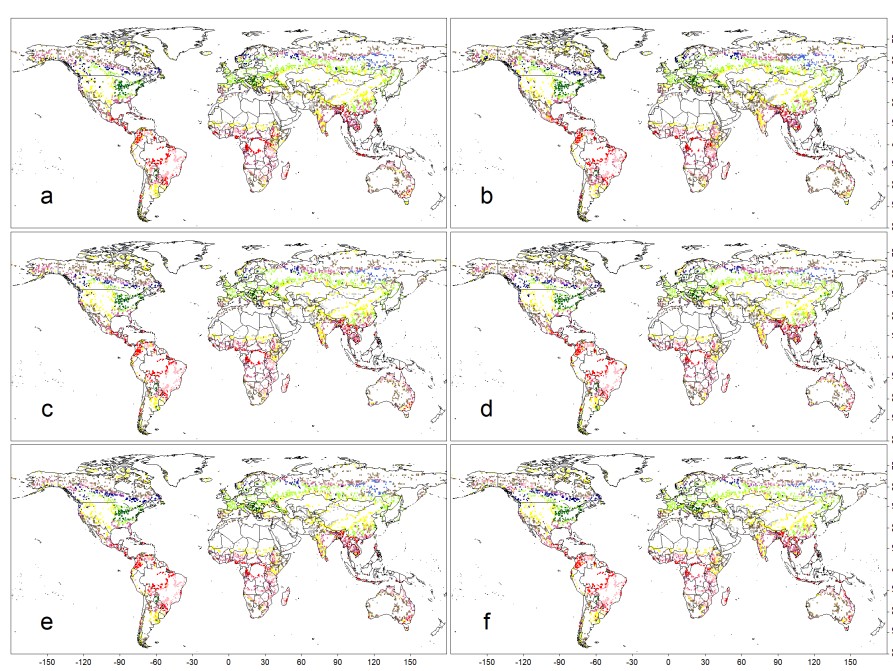

Figure 2. Differences in simulated PNV under the current climate between a random forest (RF) algorithm-based model and PNV observation data. Four sets of climate data were used for training and simulation: (a) averaged monthly air temperature and precipitation (*Ave*), (b) averaged monthly climate indices (*AveI*), (c) Ave + climate extreme indices (*CEI*), (d) *AveI* + *CEI*, (e) *Ave* + a subset of *CEI* (*CEI_{part}*), and (f) *AveI* + *CEI_{part}*. Color definitions are available in Fig. 1.





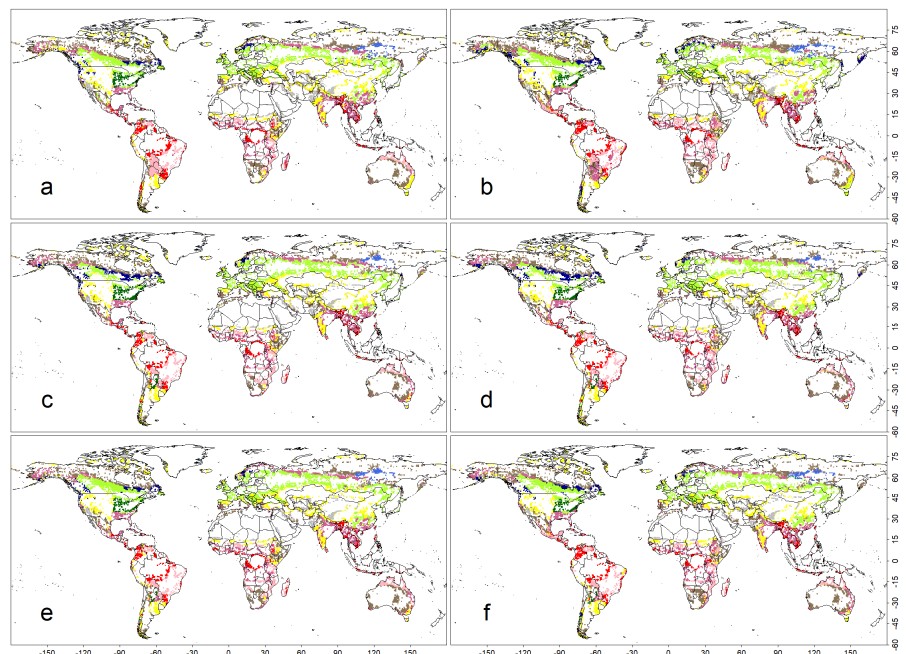


Figure 3. As in Fig.2, but for a support vector machine (SVM)-based model.




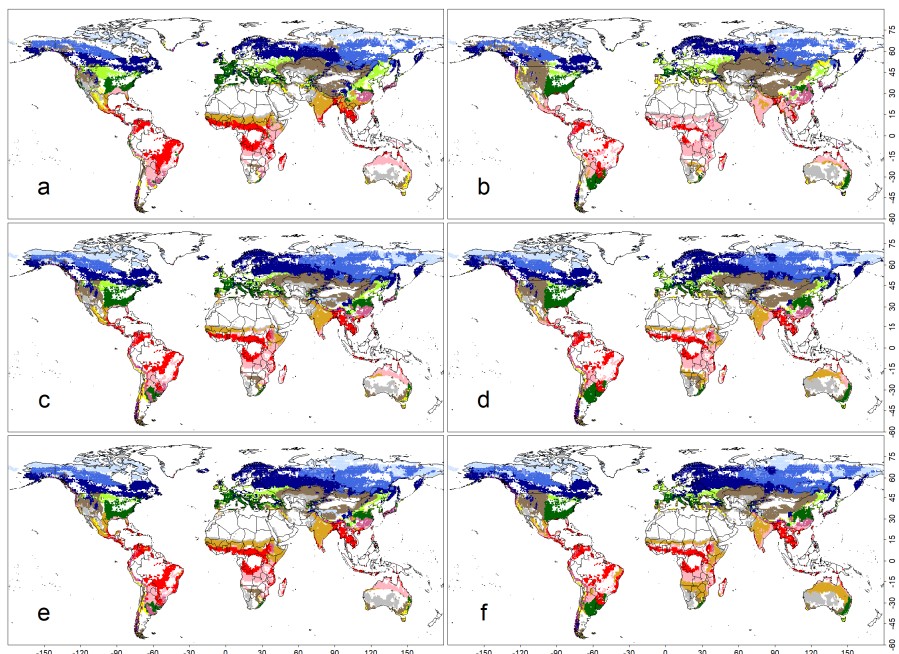


Figure 4. As in Fig. 2, but for a naive Bayes classifier (NV)-based model.






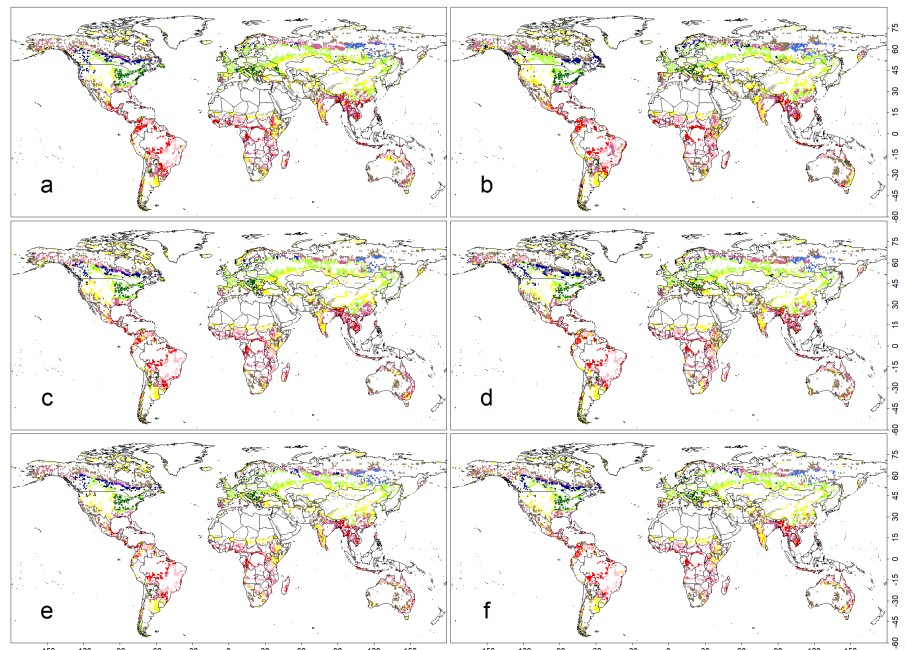

Figure 5. As in Fig. 2, but for a convolutional neural network (CNN)-based model.



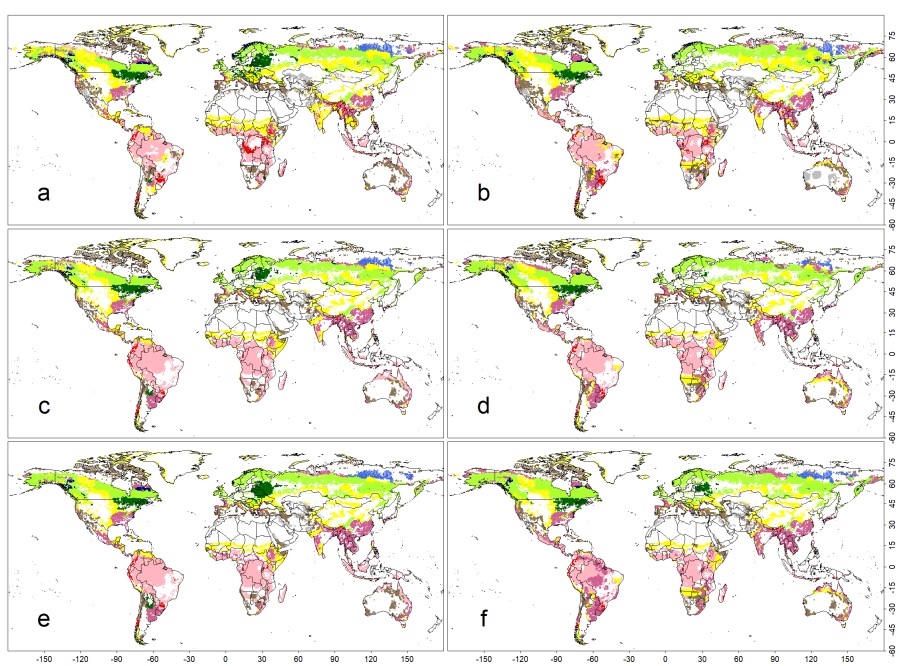

Figure 6. Differences in simulated PNV between the current climate and a Representative Concentration Pathway 8.5 (RCP8.5) climate scenario for 2080 produced by a random forest (RF)-based model. Four sets of climate data were used for training and simulation: (a) averaged monthly air temperature and precipitation (*Ave*), (b) averaged monthly climate indices (*AveI*), (c) *Ave* + climate extreme indices (*CEI*), (d) *AveI* + *CEI*, (e) *Ave* + a subset of *CEI* (*CEI_part*), and (f) *AveI* + *CEI_part*. Color definitions are available in Fig. 1.


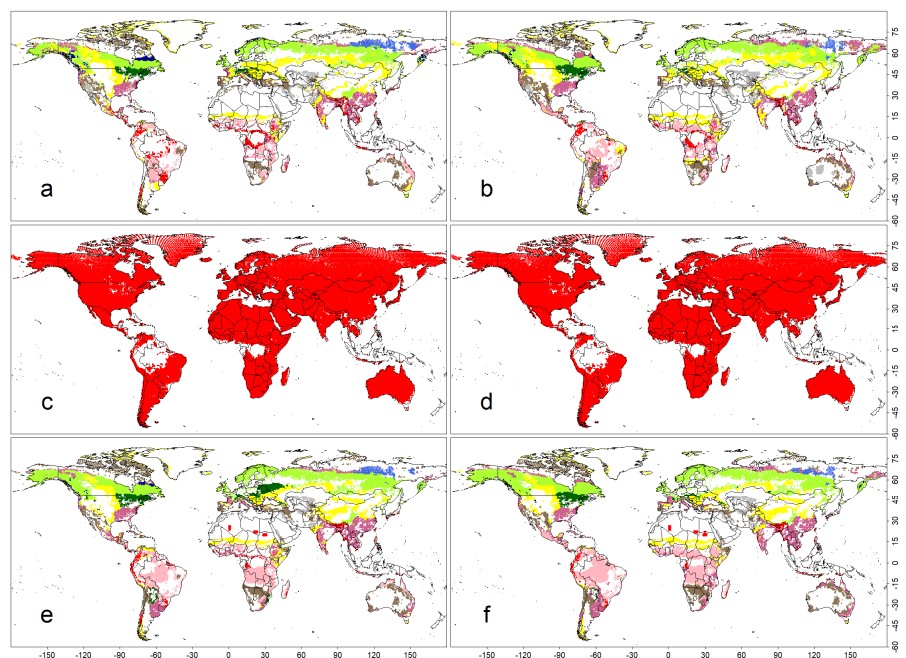


Figure 7. As in Fig.6, but for a support vector machine (SVM)-based model.





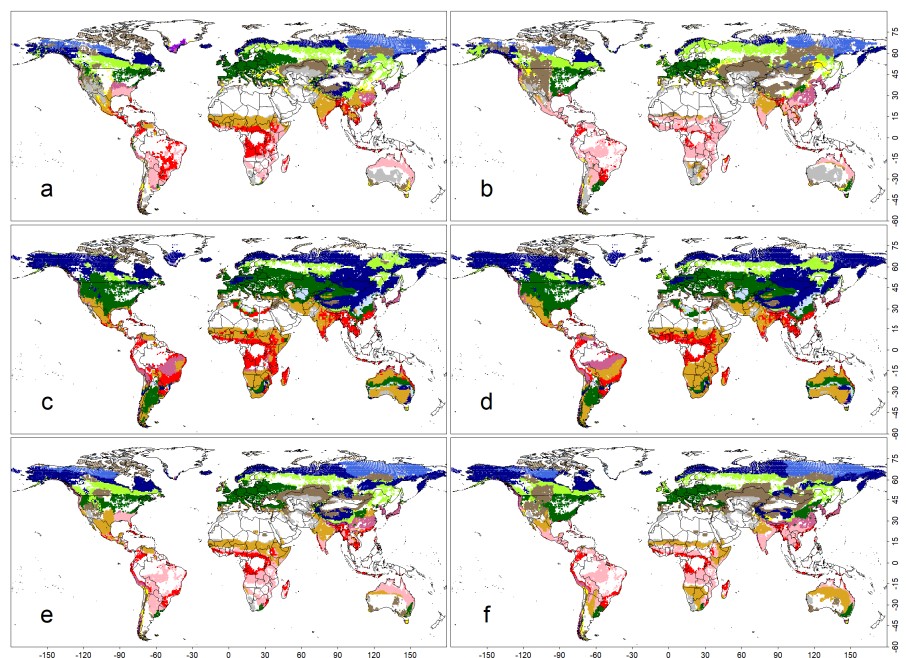


Figure 8. As in Fig.6 but for a naive Bayesian classifier (NV)-based model.






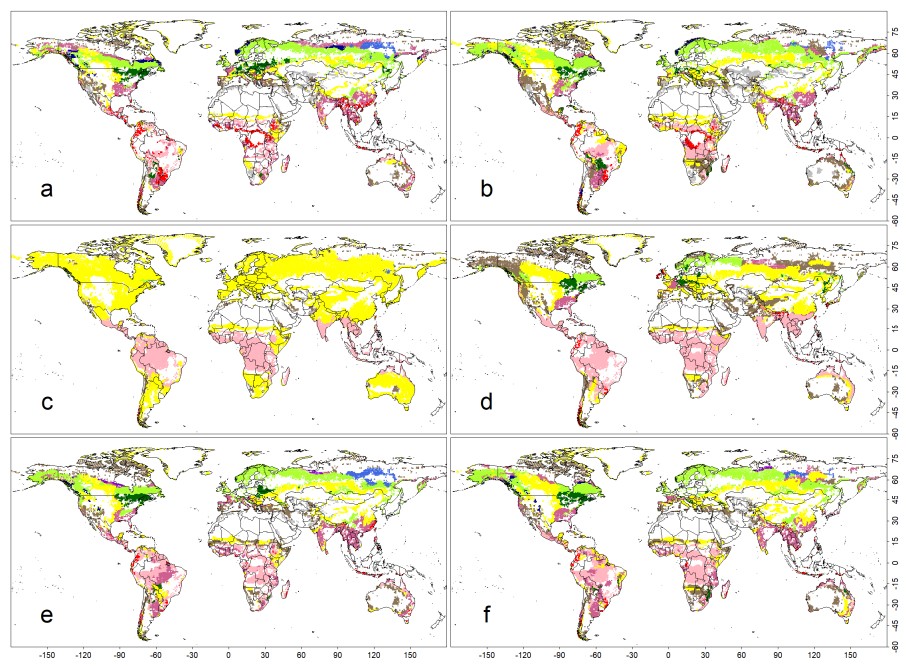

Figure 9. As in Fig.6, but for a convolutional neural network (CNN)-based model.