# Peer review of "Predicting dominant terrestrial biomes at a global scale using machine learning algorithms, climate variable indices, and extreme climate indices"

_Biogeosciences, 2023_

## Author Comment (AC1)

Dear Review #1,

Thank you very much for conducting a comprehensive review. I have addressed the issues raised to the best of my ability. Throughout this letter, given words are written in blue and numbered consecutively.

The author presents an ML study for modeling land-use based on climate data. ML models are used out of the box, trained, and then evaluated against land-use data. Land-cover predictions based on future climate are also calculated. Overall, I feel that the current manuscript is underdeveloped and would benefit from more careful model setup and validation.

General comments
(1) Model setup: This work used several ML models out of the box, without parameter refinement, which I think severely limits the value of the derived conclusions. I would generally expect some careful and thoughtful testing of different models using parameter grid search or similar, to make sure that models are indeed used to their capability. For example, the RF model is severely overfitted (100 % training accuracy), which is avoidable by limiting tree depths, etc. Hence the conclusion that CNN is preferable is not really supported here.

**Response 1:**
Parameter tuning is primarily conducted to improve training accuracy; thus, the RF model can be regarded as to be perfectly tuned in this respect. Targeting to reduce the difference between training accuracy and test accuracy (which I define as the overfitting score), adjusting the RF model by limiting the number of branches would entail tuning the model while referring to the test data. This would compromise the experimental design by not keeping calibration runs and test runs independently. In the revised manuscript, I will add an explanation on this point.

The following sentence will be inserted on Line 220:
"While the degree of overfitting of the RF algorithm could be potentially ameliorated by limiting the tree depths, doing so would involve adjusting the model by referring to the test data, thereby compromising the experimental design by not keeping calibration runs and test runs independent."

Moreover, optimizing other machine learning models is also considered impractical due to the significant amount of effort required. For instance, in the study by Sato & Ise (2022), which

serves as the basis for this paper, optimizing and conducting sensitivity analysis of a single method, CNN, required adding 19 pages of Supplementary Information. Besides, optimization depends on data-sets for training. The purpose of our paper is to provide a quick perspective on the differences in performance and uncertainties among chosen methods when non-experts in machine learning construct models for bioclimatic envelope construction using default settings. However, given that I employed each algorithm with its default settings, it precludes a definitive endorsement of CNN as the superior method.

To address this point, I will insert the following clarification on Line 207:

"While it is necessary to consider that default parameter settings are used four all methods adopted in this study, "

(2) Feature selection: The author states that because of ML, there is little necessity in limiting the number of climate indices to model land-cover. However, a large number of the climate indices that are being used in this study are highly correlated and add little additional information. In the case of the Naive Bayes that is likely a problem, because it assumes independence of features, while random forests and SVM are less affected by this. It would have been useful to determine feature importance for the ML models. Right now, we are mostly talking about black boxes, for which we don't know what features were most predictive and what causes the models to fail in the edge cases.

**Response 2:**

As noted, all climate variables are strongly correlated; for instance, the mean annual temperature and precipitation, the most basic climate variables, are positively correlated. Therefore, in our ML models, reducing the monthly mean temperature and monthly precipitation (24 variables) to a climate index dataset (16 variables) only resulted in a maximum accuracy reduction of 2%, which is one of our findings (Lines 229-232). However, if it is acceptable for the model to be a black box, such data reduction does not particularly benefit, which is the argument of this paper (Lines 232-234). Additionally, I created multiple feature sets in this paper to clarify which features each machine learning algorithm relies on and compared their performances.

I will insert the following sentence on Line 149:

"This way can avoid complexity in interpreting the results from a comprehensive examination of which feature combinations each machine learning process most strongly depends on."

Furthermore, as you pointed out, the Naive Bayes method's lower performance in my analysis might be due to the feature independence assumption.

To add this point, I will insert the following sentence on Line 171:

"This poor performance of the Naive Bayes method may be attributed to its assumption that individual features are independent of each other, a condition that climate variables cannot satisfy."

(3) Model Validation: The manuscript lumps model validation into simple accuracy scores and presents maps of deviations. It is not clear to me what the maps show (the true land-use or the false ML land use). It is also difficult to integrate over the different classes. Here something like a confusion matrix would be helpful. Generally speaking, additional validation of the ML models. Another useful validation exercise would be to compare results to a most simple model (i.e. a classical P-T relationship).

**Response 3-1:**

Figure 2-5 show maps of false biomes of ML classifications. To clarify it, I revised caption "Differences in simulated PNV under the current climate between a random forest (RF) algorithm-based model and PNV observation data" in figure 2 as "The geographical distributions of PNVs that are incorrect when compared to observation-based data in a model output based on a Random Forest (RF) algorithm."

**Response 3-2:**

I agree with your point that the model validation based on the maps in Figures 2-9 is insufficient, as well as the suggestion that confusion matrices would be helpful for a deeper interpretation of the results. However, I believe it is impractical to publish and discuss confusion matrices for all 24 models resulting from the combination of four types of ML algorithms and six datasets (4×6=24). Therefore, excluding NV for reasons explained in Lines 166-172, I have added the Confusion Matrices of models trained with the most basic AVE dataset for the remaining three models as new Table SI 2-4. Additionally, in the same table, I have included the accuracy rates for each biome class.

Table SI 2.

Confusion matrix for biome classification with AVE climate data set. Columns represent the true classes, while rows represent the RF model prediction. The shaded cells along the upper-left to lower-right diagonal represent correct classifications. For each cell, the upper line indicates the number of simulation grids, while the lower line indicates its percentage within the column, which in turn indicates the fraction of correct classifications of the corresponding biome class.

**Predicted Class**

|  | A | B | C | D | E | F | G | H | I | J | K | L | M |
|---|---|---|---|---|---|---|---|---|---|---|---|---|---|
| **A** | 958 76.2% | 0 0.0% | 0 0.0% | 8 0.6% | 149 11.9% | 0 0.0% | 3 0.2% | 51 4.1% | 9 0.7% | 76 6.0% | 1 0.1% | 2 0.2% | 0 0.0% |
| **B** | 0 0.0% | 5574 94.9% | 0 0.0% | 5 0.1% | 22 0.4% | 0 0.0% | 1 0.0% | 158 2.7% | 85 1.4% | 30 0.5% | 1 0.0% | 0 0.0% | 0 0.0% |
| **C** | 0 0.0% | 0 0.0% | 380 84.8% | 1 0.2% | 27 6.0% | 0 0.0% | 18 4.0% | 17 3.8% | 0 0.0% | 3 0.7% | 2 0.4% | 0 0.0% | 0 0.0% |
| **D** | 0 0.0% | 1 0.1% | 0 0.0% | 803 76.2% | 160 15.2% | 0 0.0% | 4 0.4% | 29 2.8% | 21 2.0% | 36 3.4% | 0 0.0% | 0 0.0% | 0 0.0% |
| **E** | 69 1.4% | 31 0.6% | 9 0.2% | 97 1.9% | 4427 88.8% | 0 0.0% | 11 0.2% | 152 3.0% | 6 0.1% | 183 3.7% | 0 0.0% | 0 0.0% | 0 0.0% |
| **F** | 0 0.0% | 1 0.9% | 0 0.0% | 1 0.9% | 6 5.4% | 26 23.4% | 24 21.6% | 17 15.3% | 12 10.8% | 22 19.8% | 1 0.9% | 0 0.0% | 1 0.9% |
| **G** | 28 0.3% | 0 0.0% | 27 0.3% | 1 0.0% | 39 0.4% | 1 0.0% | 7981 90.7% | 122 1.4% | 76 0.9% | 294 3.3% | 8 0.1% | 2 0.0% | 224 2.5% |
| **H** | 68 1.1% | 220 3.7% | 38 0.6% | 36 0.6% | 322 5.4% | 1 0.0% | 272 4.5% | 4583 76.3% | 267 4.4% | 192 3.2% | 5 0.1% | 2 0.0% | 5 0.1% |
| **I** | 14 0.3% | 270 5.5% | 8 0.2% | 5 0.1% | 20 0.4% | 0 0.0% | 134 2.8% | 272 5.6% | 4060 83.3% | 88 1.8% | 1 0.0% | 0 0.0% | 0 0.0% |
| **J** | 26 0.3% | 44 0.5% | 2 0.0% | 38 0.4% | 280 3.0% | 0 0.0% | 594 6.4% | 210 2.3% | 211 2.3% | 7770 83.3% | 0 0.0% | 31 0.3% | 125 1.3% |
| **K** | 3 1.1% | 30 10.9% | 0 0.0% | 1 0.4% | 32 11.6% | 0 0.0% | 29 10.5% | 57 20.7% | 13 4.7% | 20 7.3% | 89 32.4% | 0 0.0% | 1 0.4% |
| **L** | 5 0.4% | 0 0.0% | 0 0.0% | 1 0.1% | 1 0.1% | 0 0.0% | 5 0.4% | 2 0.2% | 0 0.0% | 68 5.7% | 0 0.0% | 1106 92.4% | 9 0.8% |
| **M** | 0 0.0% | 3 0.0% | 0 0.0% | 3 0.0% | 4 0.0% | 0 0.0% | 187 2.3% | 11 0.1% | 9 0.1% | 212 2.6% | 0 0.0% | 18 0.2% | 7636 94.5% |

(Left axis label: Predicted Class)

A: Evergreen Needleleaf Forest, B: Evergreen Broadleaf Forest, C: Deciduous Needleleaf Forest, D: Deciduous Broadleaf Forest, E: Mixed Forest, F: Closed Shrubland, G: Open Shrubland, H: Woody Savanna, I: Savanna, J: Grassland, K: Wetland, L: Snow and Ice, M: Desert

Table SI 3.

Same as table SI 2, but confusion matrix for biome classification for test accuracy with SVM Model.

**Predicted Class**

| | A | B | C | D | E | F | G | H | I | J | K | L | M |
|---|---|---|---|---|---|---|---|---|---|---|---|---|---|
| **A** | 466 37.1% | 1 0.1% | 1 0.1% | 8 0.6% | 561 44.6% | 0 0.0% | 6 0.5% | 118 9.4% | 0 0.0% | 96 7.6% | 0 0.0% | 0 0.0% | 0 0.0% |
| **B** | 0 0.0% | 5199 88.5% | 0 0.0% | 0 0.0% | 17 0.3% | 0 0.0% | 1 0.0% | 372 6.3% | 192 3.3% | 95 1.6% | 0 0.0% | 0 0.0% | 0 0.0% |
| **C** | 0 0.0% | 0 0.0% | 350 78.1% | 3 0.7% | 53 11.8% | 0 0.0% | 30 6.7% | 10 2.2% | 0 0.0% | 2 0.4% | 0 0.0% | 0 0.0% | 0 0.0% |
| **D** | 0 0.0% | 0 0.0% | 0 0.0% | 338 36.8% | 345 32.7% | 0 0.0% | 7 0.7% | 26 2.5% | 216 20.5% | 72 6.8% | 0 0.0% | 0 0.0% | 0 0.0% |
| **E** | 33 0.7% | 45 0.9% | 10 0.2% | 92 1.8% | 4140 83.0% | 0 0.0% | 21 0.4% | 347 7.0% | 9 0.2% | 288 5.8% | 0 0.0% | 0 0.0% | 0 0.0% |
| **F** | 0 0.0% | 5 4.5% | 0 0.0% | 1 0.9% | 7 6.3% | 1 0.9% | 39 35.1% | 19 17.1% | 13 11.7% | 25 22.5% | 0 0.0% | 0 0.0% | 1 0.9% |
| **G** | 50 0.6% | 1 0.0% | 55 0.6% | 1 0.0% | 81 0.9% | 0 0.0% | 7291 82.8% | 208 2.4% | 134 1.5% | 528 6.0% | 0 0.1% | 3 0.0% | 451 5.1% |
| **H** | 72 1.2% | 295 4.9% | 60 1.0% | 34 0.6% | 570 9.5% | 0 0.0% | 512 8.5% | 3556 59.2% | 583 9.7% | 323 5.4% | 0 0.0% | 0 0.0% | 5 0.1% |
| **I** | 13 0.3% | 412 8.5% | 11 0.2% | 3 0.1% | 54 1.1% | 0 0.0% | 272 5.6% | 414 8.5% | 3558 73.0% | 135 2.8% | 0 0.0% | 0 0.0% | 0 0.0% |
| **J** | 48 0.5% | 101 1.1% | 1 0.0% | 44 0.5% | 494 5.3% | 0 0.0% | 872 9.3% | 253 2.7% | 306 3.3% | 6893 73.9% | 0 0.0% | 55 0.6% | 264 2.8% |
| **K** | 14 5.1% | 39 14.2% | 0 0.0% | 1 0.4% | 56 20.4% | 0 0.0% | 60 21.8% | 71 25.8% | 11 4.4% | 22 8.0% | 0 0.0% | 0 0.0% | 1 0.4% |
| **L** | 12 1.0% | 0 0.0% | 0 0.0% | 1 0.1% | 1 0.1% | 0 0.0% | 9 0.8% | 6 0.5% | 0 0.0% | 100 8.4% | 0 0.0% | 1058 88.4% | 10 0.8% |
| **M** | 0 0.0% | 4 0.0% | 0 0.0% | 4 0.0% | 3 0.0% | 0 0.0% | 270 3.3% | 11 0.1% | 3 0.0% | 353 4.4% | 0 0.0% | 16 0.2% | 7414 91.8% |

A: Evergreen Needleleaf Forest, B: Evergreen Broadleaf Forest, C: Deciduous Needleleaf Forest, D: Deciduous Broadleaf Forest, E: Mixed Forest, F: Closed Shrubland, G: Open Shrubland, H: Woody Savanna, I: Savanna, J: Grassland, K: Wetland, L: Snow and Ice, M: Desert

*(Vertical label on left side: Predicted Class)*

Table SI 4.

Same as table SI 2, but confusion matrix for biome classification for test accuracy with the CNN Model.

**Predicted Class**

**Predicted Class**

| | A | B | C | D | E | F | G | H | I | J | K | L | M |
|---|---|---|---|---|---|---|---|---|---|---|---|---|---|
| A | 816 64.9% | 0 0.0% | 1 0.1% | 9 0.7% | 246 19.6% | 0 0.0% | 5 0.4% | 97 7.7% | 8 0.6% | 67 5.3% | 5 0.4% | 3 0.2% | 0 0.0% |
| B | 0 0.0% | 5292 90.1% | 0 0.0% | 0 0.0% | 33 0.6% | 0 0.0% | 0 0.0% | 286 4.9% | 2374 4.0% | 28 0.5% | 0 0.0% | 0 0.0% | 0 0.0% |
| C | 0 0.0% | 0 0.0% | 361 80.6% | 2 0.4% | 36 8.0% | 0 0.0% | 27 6.0% | 20 4.5% | 0 0.0% | 2 0.4% | 0 0.0% | 0 0.0% | 0 0.0% |
| D | 0 0.0% | 2 0.2% | 0 0.0% | 669 63.5% | 247 23.4% | 0 0.0% | 9 0.9% | 33 3.1% | 40 3.8% | 54 5.1% | 0 0.0% | 0 0.0% | 0 0.0% |
| E | 90 1.8% | 43 0.9% | 11 0.2% | 138 2.8% | 4127 82.8% | 0 0.0% | 14 0.3% | 256 5.1% | 6 0.1% | 300 6.0% | 0 0.0% | 0 0.0% | 0 0.0% |
| F | 0 0.0% | 5 4.5% | 0 0.0% | 1 0.9% | 8 7.2% | 15 13.5% | 26 23.4% | 20 18.0% | 15 13.5% | 20 18.0% | 0 0.0% | 0 0.0% | 1 0.9% |
| G | 49 0.6% | 0 0.0% | 53 0.6% | 2 0.0% | 51 0.6% | 2 0.0% | 7405 84.1% | 239 2.7% | 129 1.5% | 500 5.7% | 6 0.1% | 3 0.0% | 364 4.1% |
| H | 95 1.6% | 376 6.3% | 89 1.5% | 78 1.3% | 465 7.7% | 3 0.0% | 348 5.8% | 3756 62.5% | 507 8.4% | 277 4.6% | 7 0.1% | 3 0.0% | 6 0.1% |
| I | 25 0.5% | 374 7.7% | 9 0.2% | 7 0.1% | 26 0.5% | 0 0.0% | 190 3.9% | 454 9.3% | 3638 74.7% | 148 3.0% | 1 0.0% | 0 0.0% | 0 0.0% |
| J | 66 0.7% | 93 1.0% | 4 0.0% | 48 0.5% | 402 4.3% | 3 0.0% | 703 7.5% | 351 3.8% | 297 3.2% | 7039 75.4% | 0 0.0% | 56 0.6% | 269 2.9% |
| K | 9 3.3% | 37 13.5% | 0 0.0% | 3 1.1% | 41 14.9% | 0 0.0% | 44 16.0% | 85 30.9% | 9 3.3% | 24 8.7% | 22 8.0% | 0 0.0% | 1 0.4% |
| L | 8 0.7% | 0 0.0% | 0 0.0% | 1 0.1% | 1 0.1% | 0 0.0% | 8 0.7% | 4 0.3% | 0 0.0% | 75 6.3% | 0 0.0% | 1087 90.8% | 13 1.1% |
| M | 0 0.0% | 5 0.1% | 0 0.0% | 5 0.1% | 4 0.0% | 0 0.0% | 242 3.0% | 11 0.1% | 7 0.1% | 254 3.1% | 0 0.0% | 20 0.2% | 7530 93.2% |

A: Evergreen Needleleaf Forest, B: Evergreen Broadleaf Forest, C: Deciduous Needleleaf Forest, D: Deciduous Broadleaf Forest, E: Mixed Forest, F: Closed Shrubland, G: Open Shrubland, H: Woody Savanna, I: Savanna, J: Grassland, K: Wetland, L: Snow and Ice, M: Desert

Also, I will insert the following sentence between lines 172 and 173:

All models exhibited common vulnerabilities in incorrectly classifying certain biome categories. To exemplify this, confusion matrices of models trained with the most basic AVE dataset for the three models were presented in Tables SI 2-4. The biomes "Wetland" and "Closed Shrubland" consistently demonstrated particularly poor reproducibility across all models. For the former, test accuracy rates were 32.4% for RF, 0.0% for SVM, and 8.0% for CNN. For the latter, the rates were 23.4% for RF, 0.9% for SVM, and 13.5% for CNN. In both biomes and across all models, the incorrectly identified biomes were widely dispersed.

Notably, in the case of SVM, it outputs no single grid classified as the "Wetland" and only one grid classified as the "closed shrubland" over the total 52297 grids.

**Response 3-3:**

The "Classical P-T relationship" mentioned by the reviewer is presumed to refer to concepts such as Whittaker's or Holdridge's life zones. These are simple models driven solely by two independent variables: annual precipitation and annual mean temperature (or bio-temperature), which, due to their simplicity-being representable in a single figure or table-are still frequently used today. However, their accuracy is naturally low and falls outside the scope of this paper, which aims to develop a more accurate climate envelope model. Moreover, Lines 162-165 compare the accuracy of vegetation distribution based on these simple assumptions with the accuracy of the model constructed here.

(4) Purpose: I am struggling to understand the purpose of this study. If it is to evaluate the ML models, then more careful model setup and validation is desired. If it is to show predicted change of land use with climate, then importance of features for the prediction and a careful analysis of model behaviors is needed. I am trying to understand what the use case for these models is and what is being learned that can be applied elsewhere.

**Response 4:**

To clarify this point, I will insert the following paragraph between lines 243 and 244:

"As shown above, the reliability of empirical-based models cannot be guaranteed for outside training data. In contrast, process-based models can be expected to behave appropriately, even when slightly deviating from the environmental conditions covered by observational data. This is one of the reasons why many groups have proposed and developed dynamic global vegetation models (DGVMs) with greater fidelity to processes, aiming to predict biome distribution mechanistically, considering climate, soil, and the fundamentals of plant physiology and ecology (Fisher *et al.*, 2015). The expected increase in the frequency of extreme climate values in the future, which could significantly differ from the current distribution, may justify a shift from empirical-based models, like the one developed in this study, towards DGVMs. However, current DGVMs are also not a reliable option for reconstructing plant population dynamic processes at the global scale; biome map predictions under common changing climate scenarios differ significantly from state-of-the-art DGVMs (Pugh *et al.*, 2020). Hence, empirical-based models still offer an essential role to play in the approximate mapping of biomes under changing climatic conditions."

Specific comments

(5) 2.2. Climate data: I would prefer to have the variable tables moved into the main body. This makes it much easier to understand what is being used.

**Response 5:**

We will incorporate previous Tables SI2 and previous Table SI3 as new Table 1 and new Table 2, respectively, into the main body of the manuscript. Consequently, the original manuscript's Tables 1-5 will be renumbered as Table 3-7 in the revised manuscript.

(6) L 88: "Tn10p, Tx10p, Tn90p, Tx90p, WSDI, and CSDI" > need to be defined

**Response 6:**

These climatic indices are defined in Table SI3 of the previous manuscript, which is referenced immediately before the mentioned section (Line 86). Following the action taken in response to comment (5), where Table SI3 from the previous manuscript will be included in the main body as Table 2 in the revised manuscript, we believe this will eliminate any confusion for the reader.

(7) L101: Citation software artifact

**Response 7:**

Thank you for your feedback. We will make the changes in the revised version.

(8) L110: "they were transformed onto 10 min × 10 min grids through conservative interpolation and then resampled to 50 km × 50 km grids using nearest-neighbour interpolation" > I am a bit confused here, because 10x10 minutes is smaller than 50 x 50 km, so I don't understand how nearest neighbor interpolation makes sense.

**Response 8:**

The CEI dataset in question was developed by Beigaite *et al.* (2022). In section 2.1 of their paper, they explained, "The original data set had a resolution of 0.05 × 0.05 degrees. We first regridded it to 10 × 10 min grids and then resampled it to 50 × 50 km grids in line with the climatic variables used in the study." The section you have highlighted is a rephrasing of this quote. Unfortunately, their paper does not specifically mention why they chose to perform this transformation.

(9) L129: "I used the default model parameters for simplicity and to prevent potential overfitting, i.e., training the model too closely to a particular dataset, thereby creating a model that might fail to fit additional data or reliably predict future observations." > This is almost certainly a bad choice and causes an unfair comparison to the CNN.

**Response 9:**

Please refer my response to your comment (1).

**(10) L 136: (Sato and Ise, 2022): I am still a bit confused about this paper. CNNs are being used to evaluate graphical information. I still don't understand the advantage of coding numeric values in graphical information, which is then turned back into numeric information. How is this advantageous compared to for example a ANN architecture? CNNs are powerful, because photos etc. have very complex abstract features, which have to be learned by the model, but the synthetic images from climate data only have a few bits of information.**

**Response 10:**

To clarify the advantage of coding numeric values in graphical information in the CNN method, I will add explanation as followings.

Present manuscript (Lines 136-137): "I follow Sato and Ise (2022) in training our CNN with graphical images as input variables representing climatic conditions."

Revised manuscript: "I follow Sato and Ise (2022) in training our CNN. This method represents climatic conditions using graphical images and employs them as training data for CNN models. This method can automatically extract non-linear seasonal patterns for climatic variables relevant to biome classification."

**(11) L167: "The low test accuracy of the NV model was caused by an overestimation of areas dominated by boreal forest, tropical rainforest, and deciduous broadleaf forest" > generally confusion matrices or Sankey diagrams would help to understand patterns in miss-classification of the models.**

**Response 11:**

I will conduct a discussion while referring to the newly created confusion matrices. Please refer to response 3-2 above.

**(12) L202: "Excluding models trained with the NV algorithm and CEI dataset produced highly coincident PNV distributions under a future climate (accuracy, 51.7%-82.8%, Table 5)." > is 52% agreement really highly coincident. Again, it would be good to actually be presented disagreement/agreement by class rather than maps. That would actually be helpful in understanding these models.**

**Response 12:**

I will conduct a discussion while referring to the newly created accuracy reconstructed by each biome. Please refer to response 3-2 above.

(13) L235: "Adding extreme climate data improved test accuracy rates slightly but it can fatally reduce model robustness, which was defined as the consistency of model prediction under forecast climate conditions" > this is a good point, but also expected since statistical models will have a hard time making predictions/ extrapolations outside the training space. In a warming climate, extreme values will be outside what the models are trained on.

**Response 13:**

You are correct, and this is a limitation to all empirical models, which are fundamentally applicable only within the range of the training data. For this issue, please refer my response to your comment (4).

(14) Tables 4+5: I don't understand the meaning of the *. Is this just for emphasis (i.e. considered better based on author's judgement?)

**Response 14:**

I will supplement the captions in Table 4+5 as follows.

Original "Asterisks indicate the exclusion of models trained with the naive Bayes classifier or including climate extreme indices as input data"

Revised: "Asterisks indicate the exclusion of models with poor accuracies (i.e., excluding models trained with the naive Bayes classifier or including climate extreme indices as input data.)."

(15) Figures 2-9 should have color bars added.

**Response 15:**

Ok, I will!

**I will cite following references in the revised manuscript.**

Fisher, R. A., et al. (2015). "Taking off the training wheels: the properties of a dynamic vegetation model without climate envelopes, CLM4.5(ED)." Geoscientific Model Development 8(11): 3593-3619.

Pugh, T. A. M., et al. (2020). "Understanding the uncertainty in global forest carbon turnover." Biogeosciences 17: 3961-3989.

Best,
Hisashi SATO

---

## Author Comment (AC2)

Dear Reviewer #2,

Thank you very much for conducting a comprehensive review. I have addressed the issues raised to the best of my ability. Throughout this letter, given words are written in blue.

This manuscript aims to predict the global biome distribution using machine learning method based on climate characteristics and estimates its accuracy. Although CNN is a promising technology for image based vegetation classification, it remains limitedly underutilized for climate envelope modeling. I think it is good to see them being used here.

Unfortunately however, I have some concerns regarding the comparison of model performance. The authors need to clearly state that different type of input data was used to CNN and other models. The authors should also explain how these differences in data sources affected the results of the model performance comparisons.

I have four main concerns:
(1) The input data for training is not clear - The author explains that CNN uses graphical images of climate data as training data. I am not sure what graphical images of climate data meant. Is this RGB transformed climate data? If so, the authors need to clearly explain how they converted the climate data to graphical image data. The input data for RF, SVM, and NV also unclear. Are these models trained by climate data variable itself? If so, the authors also need to clearly state CNN and the other models were trained by different type of data. The authors should provide more detailed explanation about how the models used in this study trained.

**Response 1-1:**

The details of the method for converting climate data into graphical images are based on a previous publication (Sato & Ise, 2022). While this manuscript cited that paper, I will insert the following explanation on Line 137:

"The size of one graphical image is $256 \times 256$ pixels, and this image is divided into rectangular cells for as many data points as it represents, arranging tiles in each cell that express the values in grayscale. Before this visualization, climate variables were standardized to 0.01-1.00 with log transformation. The R code for drawing images is available in online open data."

If you find the above insufficient, I can also include examples of the generated images and further details of the image conversion in the supplemental information.

**Response 1-2:**

I will add an explanation as follows to clearly state that CNN and the other models were trained by different types of data.

Present manuscript (Lines 133):
"CNN algorithms are more complex than the others included in this study."

Revised manuscript:
"Although models except CNN were trained by climate data themselves, the application of CNN algorithms requires converting climate data."

Also, please refer to my response 3, which clarifies that models except CNN were trained by climate data itself. For your concern that the difference in accuracy or robustness between CNN and the other models would reflect the difference in input data, please refer to my response 4.

(2) One of my main concerns is that the fairness of model performance comparison. My understanding is that in this paper, CNN model was trained by graphical images of climate data while the other models were trained by climate data itself. Therefore, difference of accuracy or robustness among CNN and the other models seen to reflect not only model performance but also input data difference. Since model performance comparison generally aims to evaluate the performance of algorithms, I am not completely sure whether the model performance comparison in this study is truly meaningful or not. First, the author needs to clarify the reason why convert the climate data into graphical image. Second, the authors should also explain how these differences in data sources affected the results of the model performance comparisons.

Response 2-1:
CNNs are being used to evaluate graphical information. Sato & Ise (2022), the basis for this paper, developed and evaluated a method for coding numeric values in graphical information and then employed its classification with CNN. To clarify this method's advantages, I will explain as follows.

Present manuscript (Lines 136-137):
"I follow Sato and Ise (2022) in training our CNN with graphical images as input variables representing climatic conditions."

Revised manuscript:
"I follow Sato and Ise (2022) in training our CNN. This method represents climatic conditions using graphical images and employs them as training data for CNN models. This method can automatically extract non-linear seasonal patterns for climatic variables relevant to biome

classification."

**Response 2-2:**

Please refer to my response 4.

(3) P5 L129-132:  Authors should explain the model setting so that other researchers can check the validity of their methods without checking code. I feel that the descriptions of the settings of machine learning methods other than CNN are insufficient. The authors should clarify more detail about the settings of machine learning methods, such as the information of the parameters or the type of kernel they employed.

**Response 3**

Sure! I will supplement the sentence in lines 129-131 as follows.

Previous:

"I used the default model parameters for simplicity and to prevent potential overfitting, i.e., training the model too closely to a particular dataset, thereby creating a model that might fail to fit additional data or reliably predict future observations."

Revised:

"More specifically, I utilized the commands $randomForest(VegNo\sim., \; Dataset\_Train)$, $ksvm(VegNo\sim., \; Dataset\_Train)$, and $naiveBayes(VegNo\sim., \; Dataset\_Train)$, where $Dataset\_Train$ represents the training dataset table, and $VegNo$ is the name of the column within the table that holds the biome category. By opting for the default settings in these commands, I aimed to maintain simplicity and mitigate potential overfitting. Overfitting occurs when a model is trained too closely to a specific dataset, leading to a model that may perform poorly on new data or reliably predict future observations.

(4) P5 L129-132: I also have concern regarding the parameter optimization of ML. In this study, author used ML without optimizing parameters. I would suggest that the author try to optimizing parameters of ML using commonly used method such as grid search.

**Response 4**

Optimizing machine learning models requires significant amount of effort required. For instance, in the study by Sato & Ise (2022), which serves as the basis for this paper, optimizing and conducting sensitivity analysis of a single method, CNN, required adding 19 pages of Supplementary Information. Besides, optimization depends on data-sets for training. The purpose of our paper is to provide a quick perspective on the differences in performance and uncertainties among chosen methods when non-experts in machine learning construct models

for bioclimatic envelope construction using default settings. However, given that I employed each algorithm with its default settings, it prevents us from conclusively stating that CNN is the superior approach.

To address this point, I will insert the following clarification on Line 207:
"While it is necessary to consider that (1) default parameter settings are used for all methods adopted in this study, and (2) models except for CNN were trained with climate data themselves, while CNN employed graphically converted climate data, it prevents us from conclusively stating that which is the superior approach. However, "

(5) Minor points
P4 L101: Please check the format of references
**Response 5**
Sure! Thanks for noticing it.

Best,
Hisashi SATO